# A Narrative Review on the Neuroprotective Potential of Brown Macroalgae in Alzheimer’s Disease

**DOI:** 10.3390/nu16244394

**Published:** 2024-12-20

**Authors:** Melis Cokdinleyen, Luana Cristina dos Santos, Cristiano José de Andrade, Huseyin Kara, Nieves R. Colás-Ruiz, Elena Ibañez, Alejandro Cifuentes

**Affiliations:** 1Foodomics Lab, Institute of Food Science Research (CIAL, CSIC), 28049 Madrid, Spain; melis.cokdinleyen@cial.uam-csic.es (M.C.); luana.dsantos@csic.es (L.C.d.S.); rocio.colas@csic.es (N.R.C.-R.); elena.ibanez@csic.es (E.I.); 2Chemical and Food Engineering Department (EQA), Federal University of Santa Catarina (UFSC), Florianópolis 88040-900, Brazil; 3Faculty of Sciences, Department of Chemistry, Selçuk University, Ismetpasa Cad, Selçuklu, 42250 Konya, Türkiye; hkara@selcuk.edu.tr

**Keywords:** brown macroalgae, Alzheimer’s disease, extraction techniques, characterization, neuroprotection

## Abstract

Systematic Alzheimer’s disease (AD) is a neurodegenerative disease increasingly prevalent in the aging population. AD is characterized by pathological features such as *β*-amyloid (A*β*) plaque accumulation, tau neurofibrillary tangles formation, oxidative stress, an impaired cholinergic system, and neuroinflammation. Many therapeutic drugs have been developed to slow the progression of AD by targeting these pathological mechanisms. However, synthetic drugs, such as donepezil and memantine, can often lead to side effects. In this context, seaweeds have been drawing attention as a nutrient source and a potential source of health-improving metabolites. Studies have shown that extracts from brown macroalgae can potentially reduce the inflammation associated with neurodegenerative diseases by inhibiting proinflammatory cytokine expression. Furthermore, their bioactive compounds exhibit antioxidant properties vital in combating oxidative stress. Antioxidants, mainly carotenoids and phenolic compounds, have been linked to improved cognitive function and a reduced risk of neurodegenerative disorders by protecting neuronal cells through their ability to scavenge free radicals. In addition, omega-3 fatty acids found in certain macroalgae have the potential to support brain health and cognitive function, further enhancing their neuroprotective effects. In conclusion, this review has comprehensively evaluated the research conducted on brown macroalgae in the last five years, covering their potential bioactive compounds, methods of obtaining these compounds, and their neuroprotective properties against AD. The limited number of clinical studies in the literature highlights the need for further research. This narrative review provides a basic framework for new approaches to neuroprotective strategies, such as those associated with brown macroalgae natural resources. Furthermore, they may play an increasingly important role in developing functional foods and nutraceuticals that can support human health in preventing and managing neurodegenerative diseases.

## 1. Introduction

Demographic change towards an aging population is one of the main reasons for the increase in neurodegenerative diseases, especially Alzheimer’s disease (AD). Currently, approximately 35 million people worldwide are affected by AD, and this number is estimated to increase to 65 million by 2030 [1]. AD is a complex neurodegenerative disease due to its multifactorial nature, including amyloid-beta (A*β*) accumulation, tau protein hyperphosphorylation, neuroinflammation, and oxidative stress [2]. Current treatments focus on symptomatic relief rather than addressing the underlying causes of the disease [3]. Due to the inability of this approach to completely halt the progression of the disease, there is increasing interest in alternative treatment strategies for a more effective treatment. Bioactive compounds derived from algae may offer a new generation of versatile therapeutic approaches for treating AD.

Macroalgae, commonly known as seaweed, is divided into three taxonomic groups: Rhodophyta (red), Chlorophyta (green), and Phaeophyta (brown). Of these, brown algae are the most consumed species (66.5%), followed by red (33%) and green (5%) algae [4]. Seaweeds live in various marine environments, from tropical reefs to polar waters, and show distinct adaptations to different habitats. Environmental factors, such as geographic origin, water temperature, salinity, and nutrient availability, significantly affect seaweeds’ biochemical composition and bioactive properties [5]. These environmental factors may alter the biochemical content of seaweeds, further diversifying the health benefits and potential bioactive compounds of species with the generally recognized as safe (GRAS) status. Species such as *Fucus vesiculosus*, *Fucus serratus*, *Himanthalia elongata*, *Undaria pinnatifida*, *Laminaria digitata*, *Laminaria saccharina*, and *Laminaria japonica* are considered safe for human consumption, and *Undaria pinnatifida* (wakame) and *Laminaria japonica* (kombu) in particular have GRAS status. These are a source of sustainable diets and require minimal resources to cultivate when compared with terrestrial crops, making macroalgae an environmentally friendly alternative for producing functional products rich in bioactive substances [6]. As they are rich in various bioactive compounds, including polyphenols [7], polysaccharides [8], carotenoids [9], and omega-3 fatty acids [10], brown algae play a role in regulating neuroinflammatory pathways, inhibiting cholinesterase, and thus protecting against the neurodegenerative processes associated with aging (Figure 1).

The initial focus of the thorough evaluation of the potential of these compounds was the extraction and investigation of bioactive compounds from macroalgae. Efficient extraction of bioactive compounds from macroalgae depends on the type and efficiency of the extraction methods used. In this direction, the combined evaluation of traditional methods and modern techniques is essential in improving the quality and functionality of the products obtained. In particular, advanced extraction techniques, such as enzyme-assisted extraction and supercritical fluid extraction, are used to maximize the yield and bioactivity of these products [11,12]. Efficient extraction and utilization of these bioactive products can significantly contribute to the development of new dietary supplements to prevent neurodegenerative diseases [13]. Due to the variability in the bioactive organic contents of different macroalgae species, the effects of these compounds on neurodegenerative diseases should be carefully evaluated [14]. Moreover, promising macroalgae compounds for neuroprotection should be assessed with dosage and formulation tests in order to minimize their potential risks and maximize their benefits.

Research is still ongoing to understand the deficiencies in neuroprotective mechanisms. Thanks to their rich nutrient content, macroalgae are expected to play an increasingly important role in developing functional foods and dietary supplements that support brain health. This narrative review aims to reveal the potential of bioactive compounds found in brown macroalgae as a source of neuroprotection.

## 2. Neuroprotective Compound Extraction from Brown Seaweed

Natural compounds with neuroprotective properties against AD are being investigated intensively because effective treatment options are limited and because such compounds generally have low side effect profile activities. Lignans, flavonoids, tannins, polyphenols, triterpenes, sterols, and alkaloids obtained from different natural sources have attracted attention as anti-AD compounds due to their antioxidant, anti-amyloidogenic, anticholinesterase, and anti-inflammatory properties [3]. Developing effective extraction methods is critical when seeking to benefit from the potential therapeutic effects of these natural compounds. Extraction techniques are decisive in obtaining these bioactive compounds efficiently, preserving their structures, and optimizing their biological activities. For instance, conventional extraction methods (solid–liquid extractions (SLEs)) are most commonly applied due to the simple configuration of the equipment and the readily available material required. However, such extractions usually employ highly toxic solvents (e.g., hexane, hydrochloric acid, chloroform, etc.) [15,16] and could be more efficient regarding extraction yield, solvent consumption, and extraction times. Therefore, emerging technologies such as ultrasound-assisted extraction (UAE), supercritical fluid extraction (SFE), pressurized liquid extraction (PLE), and subcritical water extraction (SWE), among others, have been recently described in the literature as greener alternatives to improve the efficient recovery of compounds with neuroprotective activity [17,18,19]. Details of such applications for several brown seaweed species are described in the following sections.

### 2.1. Conventional Solid–Liquid Extraction

Though eco-friendly ways to extract active substances from matrixes are needed, simple methods like SLE are still popular because they are ready and provide helpful information about seaweed extracts.

Brown algae species can be a rich source of carotenoids such as fucoxanthin [20] and other specific compounds that are mainly found in brown seaweed, for instance, phlorotannins [21] and fucoidans [22,23], that present potential against AD. The SLE methods mostly applied to obtain seaweed extracts are soaking (also reported as stirring) or Soxhlet extraction. Both methods rely on the solvent selection, that is, in the affinity with the solutes. Soaking involves placing the ground or powdered material in the same flask as the solvent for enough time to transfer the solutes from the solid matrix to the solvent [24]. The sample is placed into a cotton thimble in a chamber in the Soxhlet extraction. The solvent (e.g., petroleum ether, hexane, ethanol, etc.) is then boiled, thus flowing through the sample several times (facilitated by a continuous condenser), and the extractable compounds are finally collected in the bulk liquid [25].

Fucoidans from *Sargassum angustifolium* have been studied as potential cholinesterase inhibitors [16]. The authors used pretreatment depigmentation of the powder material before extracting fucoidans under acidic conditions using HCl 0.1 M, stirring at 250 rpm at 60 °C for 2 h. Further steps were applied to separate the fucoidans fraction. Ethanol was mixed with the extract, and the mixture was cooled for 12 h before fucoidans were resuspended in distilled water and dialyzed. Recently, another class of compounds from *Sargassum macrocarpum,* named meroterpenoids, has been described as a possible alternative path to decrease A*β* aggregation [26]. The extraction of the meroterpenoids was performed using SLE with ethanol, followed by purification steps using hexane, methanol, and ethyl acetate as solvents. About 16 meroterpenoids were isolated from the brown algae [26].

Shrestha et al. (2021) focused on a unique class of phenolics from *Ecklonia radiata*, phlorotannins, particularly dibenzodioxin-fucodiphloroethol. To obtain an enriched extract in this phlorotannin, authors applied SLE followed by purification steps [27]. The global extract was obtained with ethanol, while a subsequent refined extraction to obtain the purified compound was achieved by applying SLE with water, ethyl acetate, and butanol. High-performance counter-current chromatography (HPCCC) was also used to extract high-purity content in the target compound.

Recently, Martens et al. (2023) evaluated extracts from seven brown seaweeds (*Alaria esculenta*, *Ascophyllum nodosum*, *Fucus vesiculosus*, *Himanthalia elongata*, *Saccharina latissima*, *Sargassum muticum*, and *Sargassum fusiforme*) as potential neuroprotectors [28]. The extraction procedure involved soaking the dried seaweed samples in a chloroform:methanol mixture (2:1) at room temperature and under UV light exposure overnight. Then, 10 min sonication followed by paper filtering was applied to the rota-evaporated mixture to obtain the lipid extract. For the first of these, the most efficient activation was achieved using the extracts from H. elongata, which presented the highest saringosterol content. In contrast, the best activation results for the latter stood for *A. esculenta*, *A. nosodum*, *F. vesiculosus*, *S. Latissima*, and *S. muticum*.

### 2.2. Advanced Extraction Technologies for Brown Macroalgae Neuroprotection: A Greener Perspective

Traditional extraction methods use harmful solvents, but new techniques like supercritical CO_2_ extraction, subcritical water extraction (SWE), pressurized liquid extraction (PLE), ultrasound-assisted extraction (UAE), enzymatic-assisted extraction (EAE), and microwave-assisted extraction (MAE), use safer options and improve the process, making them better choices for extracting valuable compounds. For instance, PLE comprises a wide range of compound polarity as many solvents can be used, and it usually operates at temperatures above the boiling point of the solvent, which is one of the main advantages of this technique [29]. Similarly, in SWE, the extraction occurs at high temperatures, with water as the extraction solvent [30]. Likewise, SWE and EAE are usually associated with some hydrolysis processes and are thus applied when the target is a smaller fraction of some compound such as peptides, monosaccharides, etc. [31,32].

Non-pressurized techniques, such as UAE, usually intensify the extraction process through cavitation phenomena that create, expand, and implode the formed microbubbles in the system. These phenomena favor the extraction of compounds when they occur close enough to the solid matrix [33]. Such technologies support the release of the compounds from the matrix to the solvent. The following examples associate these new methods with compounds from brown algae with neuroprotective activity.

Hans et al. (2024) performed SWE to recover biopeptides from *Padina tetrastromatica* sequentially after fucoidan extraction [32]. The extraction temperatures ranged from 100 to 220 °C, and the pressure was fixed at 400 bar. The brown algae biomass to water ratio was 1:50 (*w*/*v*) and stirring of the system was kept at 250 rpm. The hydrolysis was stopped by cooling the system at the end of the reaction. The authors also performed conventional extraction using an aqueous solution (pH = 11) incubated at 30 °C for 45 min. The protein and free amino acid contents from the hydrolysates obtained from SWE was up to 80% higher than conventional extraction, and their recovery was higher as the temperature increased. These peptides, very likely obtained at 220 °C, exerted positive effects against AChE activity, achieving an IC_50_ value of 17.9 mg mL^−1^ against 65.9 mg mL^−1^ from conventional extraction. It is worth noting that the by-product from fucoidan production was employed to obtain biopeptides; therefore, the approach is an excellent alternative when seeking to simultaneously obtain products with different properties.

Soares et al. (2021) also used the ScW process to obtain extracts from two different brown algae: *Fucus vesiculosus* L. and *Codium tomentosum* [34]. Unlike the previously mentioned work, the pressure was 100 bar instead of 400 bar. The authors implemented the extraction in a continuous operational mode, under 10 mL min^−1^ water flow. They obtained four samples with temperatures ranging from room temperature to 250 °C. The activity in the enzyme inhibitions of the respective fractions were investigated for AChE, BChE, and tyrosinase, all brain enzymes that are associated with neurodegenerative diseases. Similar to the results obtained from Hans et al. (2024) [32], the result of enzyme inhibition showed positive effects when using higher extraction temperatures, as the last fraction (190–250 °C) showed the best inhibition effect for all enzymes tested.

High-pressured liquids (water and others) were also used as solvents to recover bioactives from brown algae. In the example of Ruiz-Domínguez et al. (2023), PLE was applied to *Durvillaea antarctica* (popularly known as cochayuyo) [19]. Firstly, the effects of different solvents (water, water/ethanol 50:0 *v*/*v*, ethanol, ethyl acetate, acetone and heptane) in the global yield and antioxidant capacities (DPPH and TEAC) obtained by conventional extraction were evaluated. As a further step, for the best solvents (water, water/ethanol), a pressurized liquid extraction system (static) was set to 10.34 bar and 30 min, with temperatures varying from 40 to 180 °C and ethanol concentration from 0 to 100% (v v^−1^), where the best AChE result (IC_50_ = 148.62 µg mL^−1^) was found for 110 °C and pure water. When 100% ethanol was used, AChE consistently presented the worst result (IC_50_ = 569.05 µg mL^−1^), suggesting a possible correlation of enzyme inhibition with the concentration of more hydrophilic compounds. However, considering all effects in the responses (yield, antioxidant capacity and AChE inhibition), the best extraction condition was obtained at near to 180 °C, confirming brown algae extracts’ tendency to achieve enhanced enzyme inhibition results at higher temperatures.

On the other hand, in their targeted extraction of polyphenols, Jiang et al. (2024) applied ultrasound-assisted extraction (UAE, 40 kHz, 500 W) to *Sargassum pallidum* [18]. The optimized condition was as follows: ethanol = 50%, ultrasonic temperature = 60 °C, and total extraction time = 40 min. The authors tested the inhibition activity of the extracts against the enzymes α-amylase, α-glucosidase, tyrosinase, and AChE. Although positive controls (acarbose, kojic acid, and donepezil) demonstrated more substantial inhibitory potential, excellent results were achieved for α-glucosidase (IC_50_ = 6.290 μg mL^−1^), suggesting a potential alternative and indirect treatment for AD by regulating blood glucose [35].

To obtain a fucoxanthin-enriched extract from *Sargassum* species, Hong et al. (2023) used an alternative method (acetone extraction followed by ultrasonication) to obtain a highly purified fraction of fucoxanthin for *Sargassum oligocystum* (from 0.4 to 3.0 mg g^−1^, depending on the sample collection season) [20]. The extracts resulted in promising neuroprotective effects, confirmed by the absence of cytotoxicity and protection against induced neuronal death.

Boucelkha et al. (2017) have described an EAE method by which to access the oligoguluronates from *Stypocaulon scoparium* using the enzyme alginate lyase [31]. Carbohydrate hydrolysis was achieved more sustainably than other conventional methods, such as acid hydrolysis, which is often described for the recovery of these compounds [36].

Regarding saccharides from brown algae, fucoidans play a key role as they present promising neuroprotective activities [37]. The fucoidans were extracted (method unknown) from *Fucus vesiculosus* and *Undaria pinnatifida*, with the latter showing the most promising results for neuroprotective activity, as its effects were most closely related to the fucoidan molecular structure, which may vary depending on the extraction/purification method. In this sense, process intensification represents an alternative to overcome such limitations. Quitain et al. (2013) present a possible alternative by applying a multi-step extraction to enhance fucoidan recovery from *Undaria pinnatifida* [38]. The authors applied SFE as a sample defatting method before MAE hydrolysis (600 W, 5–120 min, 110–200 °C), successfully recovering low molecular weight fucoidans at 140 °C and 30 min. Such a strategy suggests an enhancement in the bioactivity against AD when applying this extract.

## 3. Chemical Characterization of Extracts from Brown Macroalgae

After extracting the target compounds, detecting and identifying the analyzed molecules is essential to characterizing and quantifying the analytes. Different analytical platforms must consider the wide range of chemical structures associated with brown algae’s neuroprotective potential. Chromatographic separation, followed by a detection instrument, is one of the most widely employed techniques for determining compounds from these natural sources. Although several chromatography techniques can be used based on the physicochemical properties of the analytes, advanced techniques, such as high-performance liquid chromatography (HPLC), are usually the most utilized for bioactive compounds. To detect and provide information about the structure of bioactive compounds, numerous technologies are available, including infrared (IR) spectroscopy, fluorescence (FL) spectroscopy, ultraviolet–visible (UV–vis) spectroscopy, nuclear magnetic resonance (NMR), ion mobility spectroscopy (IMS), Raman spectroscopy, and mass spectrometry (MS). Thus, the physicochemical properties of the target compounds are determinants for the selection of the instrument. However, MS is frequently used by the scientific community, in tandem with HPLC, to characterize and to determine the composition of the chemical structure of several neuroprotective compounds from brown algae [39].

### 3.1. Liquid Chromatography

HPLC is a rapid and powerful separation technique based on conventional column chromatography, where a mobile phase, together with the sample, is passed through a column packed with a stationary phase. This is a versatile tool as a wide range of diverse molecules (i.e., fatty acids, polysaccharides, pigments, polyphenols) can be analyzed by modifying the gradient, the stationary and/or the mobile phase. Reversed-phase (RP) HPLC is the most commonly used technique in neuroprotective compounds of brown algae. HPLC is typically coupled to a detector, enhancing its capabilities by offering precise information about the spectroscopic properties of the analytes. Although MS is commonly used for characterizing bioactive compounds from brown algae, other instruments are also used (i.e., UV–vis, FL, or diode array detector (DAD)). Research experimental works where HPLC is coupled to several detectors have been applied to determine and characterize neuroprotective compounds extracted from brown algae and are described below and in Table 1.

Martens et al. (2023) analyzed the lipidomic extract of seven brown macroalgae (*A. esculenta*, *A. nodosum*, *F. vesiculosus*, *H. elongata*, *S. latissima*, *S. muticum*, and *S. fusiform*) using HPLC coupled to high-resolution mass spectrometry (HRMS) and MS/MS [28]. The use of HRMS makes it easier to determine the molecular structure of a compound by utilizing advanced software that analyzes isotope patterns and precise *m*/*z* measurements. Additionally, generating fragments from the parent molecules (MS/MS) provides a higher confirmation level for compound identification. They tentatively identified up to 323 lipids those, 47 fatty acids, 116 phospholipids, and 160 glycerolipids.

Another tool that is widely combined with liquid chromatography (LC), especially for phenolic compounds, is UV–vis spectroscopy through UV detectors and DAD (also named photodiode array detectors (PDA)). This method allows for the real-time recording of the UV–vis absorption of target molecules [40]. Nho et al. (2020) applied HPLC–DAD to quantify dieckol at 230 nm from *Ecklonia cava* extract, obtaining 48.08 ± 0.67 mg g^−1^ [41]. Another type of molecule suitable for DAD are those of the pigments. In this sense, a screening experiment where fucoxanthin was quantified in nine brown algae of the genus *Sargassum* was performed by Hong et al. (2023) [20]. The values quantified ranged from 3.82 to 2927.98 µg g^−1^ dry weight (d.w.), *S. oligocystum* was found to be the species with the highest accumulation capacity of fucoxanthin.

Although UV and DAD detectors have been demonstrated as useful tools for quantifying bioactive compounds, this approach presents certain limitations for compound identification. For instance, reference standards must be injected with the sample to compare retention times and UV–vis spectra with the unknown analytes. Additionally, more than chromatography may be needed to resolve and effectively separate the compounds in complex mixtures, complicating the identification process. Consequently, the use of LC coupled with UV detectors or DAD and MS, to provide complementary information when profiling bioactive compounds in complex matrices, is becoming more widespread [39].

An example of this can be observed in the work performed by Palaniveloo et al. (2023) [42], where the extract of *Laurencia snackeyi* was analyzed by HPLC coupled with DAD and HRMS. Thirty-seven metabolites were tentatively identified in this extract, mainly amino acids, fatty acids, terpenoids, phenolics, aromatics, and cyclic ketones (Table 1).

These analytical technologies were also used in tandem to characterize the chemical composition of the *Petalonia binghamiae* extract [43]. The major constituents identified were vachanic acid methyl ester (R)-ricinoleic acid, saikosaponin E, acanthoside D, and kansuinin D.

In very complex matrices with high chemical variability, it is possible that one-dimensional liquid chromatography does not have enough resolving power to separate the target molecules. In this case, two-dimensional (2D) LC is an interesting option, primarily when two different stationary phases are implemented, as it facilitates the characterization of the sample across multiple dimensions within a single analysis [44]. Montero et al. (2014) developed a 2D LC method for the separation and characterization of phlorotannins from brown algae *Cystoseira abies-marina* in which hydrophilic interaction liquid chromatography (HILIC) in the first dimension was combined with RP in the second dimension and coupled to DAD-MS [45]. In this work, up to 52 compounds were separated and identified, 43 of which were putatively characterized as different phlorotannins.

### 3.2. Other Techniques

Although LC is the most used technique to separate or quantify neuroprotection from brown algae, other techniques can also be used; for instance, gas chromatography–mass spectrometry (GC–MS) is usually used to identify thermally stable volatile chemicals. This technique sometimes requires a derivatization step where the suitability of analytes for GC analysis is enhanced, improving the sensitivity and peak separation [46]. According to Martens et al. (2023), the levels of saringosterol and fucosterol were measured using gas chromatography coupled with mass spectrometry (GC–MS) in seven species of brown algae. The concentrations of saringosterol ranged between 2 to 34 ng mg^−1^ (d.w.), while fucosterol concentrations ranged from 37 to 771 ng mg^−1^ (d. w.) along the analyzed species [28].

As an alternative to MS, Fourier transform infrared spectroscopy (FT-IR) detects compounds that can absorb infrared light. This is widely used to determine phenolics and other compounds, such as fatty acids [47]. Hang et al. (2024) used FT-IR to determine the composition of sulfated polysaccharides in the extract of *Sargassum horneri*. The authors putatively identified the fucoidan [47].

NMR is a valuable technique, especially for identifying pure compounds. Compared with the other detectors, one of the most significant disadvantages of NMR is that it requires several milligrams of high-purity sample. Nevertheless, NMR can give a general indication of the overall structure of the analyzed extracts [48]. Palaniveloo et al. (2023) performed an example of an NMR procedure on an extract from *L. snackeyi*. The author complemented LCMS with NMR and examined six metabolites, characterizing three of them as Palisadin A, Aplysistatin, and 5-Acetoxypalisadin B. The other three isolated compounds could not be identified due to the low yield [42].

**Table 1 nutrients-16-04394-t001:** Analytical methods for the chemical characterization of various brown algae extracts.

Family	Compounds	Brown Algae Species	Separation Technique—Detection Instrument (GC/LC Column)	Stationary Phase and Mobile Phases	Ionization Source	References
Lipids	Fatty acids, phospholipid, glyceroglycolipid, diglyceride and triglyceride.	*Alaria esculenta*, *Ascophyllum nodosum*, *Fucus vesiculosus*, *Himanthalia elongata*, *Saccharina latissima*, *Sargassum muticum*, and *Sargassum fusiforme*.	HPLC–Orbitrap–MS/MS (Poreshell 120 EC-C18)	RP. A: ACN/water 60:40 (0.1% FA and 10 mM NH_4_HCO_2_); B: IPA/ACN 90:10 (0.1% FA and 10 mM NH_4_HCO_2_).	ESI+; ESI-	[28]
Lipids	Sterols (saringosterol and fucosterol).		GC–MS (DB-XLB 122-1232)	-	EI	
Polyphenol	Dieckol.	*Ecklonia cava*.	HPLC–DAD (C18 Kromasil 100-5)	RP. A: water; B: MeOH.	-	[41]
Amino acids, fatty acids, terpenoids, phenolics, aromatics, and cyclic ketones.	6-aminocaproic acid, carnitine, 2-decenoic acid, docosahexaenoic acid, 5,8,11,14-eicosatetraynoic acid, 2,7,12,17-octadecanetetrol, 10,12-hexadecadienal, erucamide, 15-oxo-11,13-eicosadienoic acid, 16-hydroxypalmitic acid, 2-hydroxymyristic acid, 9,12,13-trihydroxy-15-octadecenoic acid, oleic acid, palmitic acid, 3,5-dibromo-4-hydroxybenzoic acid, 11-(2-hexyl-5-hydroxyphenoxy)-N- (2-hydroxyethyl) undecanamide, 6-gingerol, paradol, decanophenone, heptanophenone, haplamine, N-benzylformamide, 15-hydroxy-1-[2-(hydroxymethyl)-1-piperidinyl] prost-13-ene-1,9-dione, 8-[3-oxo-2-(2-penten-1-yl)-1-cyclopenten- 1-yl]octanoic acid, jasmone, laurendecumenyne A, coronaridine, ar-turmerone, retinaldehyde.	*Laurencia snackeyi*.	HPLC–DAD–Orbitrap MS/MS (Phenomenex C18)	RP A: water (0.1% FA); B: ACN (0.1% FA).	ESI+; ESI-	[42]
	Palisadin A, aplysistatin and 5-acetoxypalisadin B.		NMR.			
Fatty acids, terpenoids	Vachanic acid methyl ester, (R)-ricinoleic acid, saikosaponin E, acanthoside D, and kansuinin D.	*Petalonia binghamiae*.	HPLC–DAD–QToF–MS (ZORBAX Eclipse Plus C18).	RP A: water (0.1% FA); B: CAN (0.1% FA).	ESI-	[43]
Phenolics	Total of 49 compounds: 18 phenolic acids, 22 flavonoids, 6 other polyphenols, 1 lignan and 1 stillbene.	*Phyllospora comosa*, *Ecklonia radiata*, *Durvillaea* sp., *Sargassum* sp., *Cystophora* sp.	LC–ESI–QTOF–MS/MS (Synergi Hydro-Reverse Phase 80°A).	RP A: water (0.1% FA); B: ACN/water (0.1% FA) (95:5).	ESI+; ESI-	[49]
Pigment	Fucoxanthin.	*Sargassum mucclurei* (1650.03 ± 7.10), *S. binderi* (296.07 ± 5.50), *S. polycystum* (3.82 ± 0.89), *S. duplicatum* (255.03 ± 5.71), *S. denticarpum* (24.25 ± 2.72), *S. swartzii* (217.60 ± 5.04), *S. microcystum* (161.52 ± 2.90), *S. crassifolium* (26.65 ± 2.86), *S. oligocystum* (2927.98 ± 8.01) *.	HPLC–DAD (Symmetry^®^ C18).	RP ACN/MeOH (1:9, *v*/*v*).	-	[20]
Phenolics	Phlorotannins.	*Cystoseira* *abies-marina*.	HILIC × RP–DAD–MS/MS (lichrospher diol-5; Ascentis Express C18).	HILIC A: ACN/AcOH (98:2, *v*/*v*) B: MeOH/Water/AcOH (95:3:2, *v*/*v*/*v*). RP A: water (0.1% FA); B: can.	Ion trap with ESI-	[45]
Sulfate polysaccharide	Fucoidan.	*Sargassum Horneri*.	Fourier-transform infrared spectroscopy (FT-IR).			[47]
Polysaccharide	Alginate (through the measure of monosaccharides xylose, galactose, glucose, mannose, fructose, and rhamnose).	*Padina pavonica*, *Sargassum cinereum*, *Turbinaria turbinata*, and *Dictyota dichotoma*.	HPLC–UV–vis (C18).	RP A: water/ACN (90:10, *v*/*v*); B: ACN (0.045% KH_2_PO_4_–0.05% trimethylamine).		[50]
Meroterpenoids	New sargasilols (9): (10′E)-10′-dehydroxy-11′,12′-dihydro-10′,11′-didehydro-12′-hydroperoxysargachromanol G; (10′E)-10′-dehydroxy-11′,12′-dihydro-10′,11′-dide-hydro-12′-hydroxysargachromanol G; 9′-deoxy-9′-oxo-11′,12′-dihydrosargachro-manol K; 9′-deoxysargachromanol K; 3′,4′-dihydro-4′,16′-didehydro-3′-oxosargachromanol E; 6′-hydroxy derivative of sargachromanol G; 15′-hydroxysargachromanol L; methylsargachro-manol E.	*Sargassum siliquastrum*.	NMR.			[51]

* Concentration of target compound, measured in each specie. RP: Reversed phase; ACN: acetonitrile; MeOH: methanol; AcOH: acetic acid; NH_4_HCO_2_: ammonium formate; KH_2_PO_4_: monopotassium phosphate.

## 4. Neuroprotection Assays of Brown Macroalgae Extracts

AD is a complex and multifactorial disease resulting from the combination of several interacting factors (Figure 2). Due to this complexity, an ideal set of biological assays should be used to evaluate the potential of anti-AD compounds. These methods include the evaluation of enzyme inhibitors such as AChE and BChE and of biomarkers such as the inhibition of A*β* plaques. Therefore, in neuroprotection tests, to evaluate the effectiveness of anti-AD extracts, both AChE and BChE inhibitors and measurement of the inhibition of A*β* plaques play essential roles. These tests assess the effects of compound candidates on different aspects of AD, contributing to the development of treatment strategies.

Table 2 presents the neuroprotective activity of various brown algae extracts in vitro and in vivo studies. It summarizes the findings evaluating the potential benefits of compounds derived from algae against AD in different biological models.

### 4.1. Cholinesterase Inhibitors

AChE and BChE inhibitor tests are generally in vitro assays for assessing neuroprotection. AChE is an enzyme that plays a critical role in neurotransmission and is responsible for the breakdown of a neurotransmitter called acetylcholine. Similarly, BChE is responsible for the degradation of butyrylcholine, a different substrate [52]. In diseases such as Alzheimer’s disease, excessive activity of AChE can, in particular, cause communication disorders between nerve cells, which may aggravate the symptoms of the disease. Therefore, AChE and BChE inhibitor tests play a critical role in discovering and developing potential therapeutics.

Various brown algae metabolites that inhibit cholinesterase activity are reported in Table 2. For example, Pedro et al. (2021) found that extracts of *Undaria pinnatifida* inhibited AChE enzyme activity by 50% at a concentration of 1 mg mL^−1^ [53]. In the study of Gomes et al. (2022), the neuroprotective potential of extracts obtained from *Himanthalia elongata* and *Eisenia bicycle* species was investigated. In the study, it was shown that a fraction of *H. elongata* and *E. bicyclis* (at a concentration of 2 mg mL^−1^) provided 40% AChE inhibition for *H. elongata*, 50% for *E. bicyclis*, and 40% BChE inhibition in both species. Although no high effect was observed regarding enzyme inhibition, it was determined that the fourth fraction obtained at the highest temperature (250 °C) showed particularly strong radical scavenging activities [54]. This was because the fractions obtained at high temperatures consisted of phenolic compounds and Maillard reaction products. This combination contributed to the antioxidant activities of the extracts and thus their neuroprotective properties. Ruiz-Domínguez et al. (2023) showed that the most effective extracts in terms of AChE inhibitory capacity (IC_50_ value was 148.62 μg mL^−1^) could be obtained at low-medium temperatures (40–110 °C) and using 100% water by pressurized liquid extraction (PLE) from *Durvillaea antarctica* [19].

Fucoxanthin is a naturally occurring carotenoid in brown macroalgae, and its AChE inhibition activity means that it is capable of promising neuroprotective effects. Fucoxanthin isolated from *Sargassum oligocystum* has also showed inhibitory activity on AChE (IC_50_ = 130.12 ± 6.65 µg mL^−1^) [20]. Hong et al. (2023) studied the biological activities of fucoxanthin obtained from the seaweed *Sargassum oligocystum* with acetone using sonication at room temperature. After being isolated from the seaweed *S. oligocystum*, fucoxanthin showed inhibitory activity on AChE (IC_50_ = 130.12 ± 6.65 µg mL^−1^) [20].

Brown macroalgae are rich natural sources of phenolic compounds and can inhibit cholinesterase activity. For instance, polyphenol-rich extract (PEEC) from *Ecklonia cava* was shown to have a strong inhibitory effect on AChE and BChE enzymes. In addition, the IC_50_ value of PEEC on AChE inhibition was determined as 68.9 μg ml^−1^, and the IC_50_ value on BChE inhibition was determined as 217.7 μg ml^−1^ [41]. Shrestha et al., (2021) showed that dibenzodioxin-fucodiphloroethol (DFD) from *Ecklonia radiata*, had moderate AChE inhibitory activity (IC_50_ value 41 μM) and shared similar binding residues with donepezil [27]. Park et al., (2019) showed the capacity of polyphenol extracts from *Ecklonia cava* to inhibit AChE, and the IC_50_ value was determined as 70.63 μg mL^−1^. In addition, the fucoidan extract in the study effectively suppressed AChE activity, increasing Ach levels in the brain tissue in the TMT-induced cholinergic system dysfunction model [8]. Although the polyphenol-containing extract showed AChE inhibition in vitro, this effect could not be sufficiently demonstrated in the in vivo model. It was determined that fucoidan from *Sargassum angustifolium* had moderate AChE inhibitory activity and that its IC_50_ value was 1.20 µg mL^−1^ [16]. Hans et al. (2024) determined that the hydrolysate obtained from *Padina tetrastromatica* inhibits AChE in vitro, with the IC_50_ value being determined to be 17.9 ± 0.1 mg mL^−1^ [32]. In the amino acid profile of the hydrolysate, 45% essential amino acids were detected, and it was revealed that the hydrolysate also contained phenolic compounds (23.9 ± 1.4 mg GAE g^−1^) and flavonoids (1.23 ± 0.1 mg QE g^−1^). It has been reported that the extracts of *Durvillaea incurvate* obtained by the UAE method effectively inhibited AChE (IC_50_: 48.55 ± 0.021 µg mL^−1^, 51.5% inhibition) and BChE (IC_50_: 87.58 ± 0.044 µg mL^−1^, 32.8% inhibition) enzymes [55].

Sulfated polysaccharides from brown macroalgae inhibit cholinesterase enzymes, contributing to synaptic transmission and preservation of cognitive functions. For example, sulfated polysaccharides of *Sargassum horneri* showed high selectivity on AChE, and AChE inhibition was IC_50_ = 9.77 µM. These findings indicate that sulfated polysaccharides obtained from *Sargassum horneri* can be evaluated as potential cholinesterase inhibitors against AD [47].

Future studies focusing on obtaining a wider range of biologically active compounds, examining their neuroprotective effects, optimizing these techniques, and demonstrating their applicability on an industrial scale may enable the more widespread use of cholinesterase inhibitors obtained from natural sources.

Furthermore, there are studies in which the inhibitory interactions and binding mechanisms of bioactive compounds from brown algae with target enzymes are evaluated in detail through enzyme kinetics and molecular docking simulations. Palaniveloo et al. (2023) have reported that their extracts from *Laurencia snacke* (a purplish-brown algae), showed IC_50_ values of 14.45 ± 0.34 μg mL^−1^ for AChE and 39.59 ± 0.24 μg mL^−1^ for BChE [42]. In addition, in their molecular docking analysis to identify potential candidates that can inhibit AChE and BChE, 5,8,11,14-eicosatetraynoic acid and 15-hydroxy-1-[2-(hydroxymethyl)-1-piperidinyl]prost-13-ene-1,9-dione were the compounds that exhibited better inhibition properties.

In another study, phlorotannins from *Ecklonia cava* were shown to have AChE inhibition, with IC_50_ values ranging from 0.9 ± 0.8 to 66.5 ± 0.4 μM, and BChE inhibition, with IC_50_ values ranging from 1.4 ± 3.8 to 25.2 ± 0.1 μM. In addition, enzyme kinetics and molecular docking simulations evaluated the inhibitory interaction and binding mechanism between these active compounds and the target enzyme, thus supporting their inhibitory potential [56].

A study using molecular dynamics simulation and binding energy calculations, using AChE as a target, identified diplorethohydroxycarmalol and phlorofucofuroeckol from brown seaweeds as potential lead compounds for neurodegenerative diseases [57]. When in vitro tests and simulations (dockings) are analyzed together, a more comprehensive perspective is obtained to understand the effects of potential drug compounds on biological targets. Using these two methods together offers numerous advantages when evaluating the effectiveness of potential drug candidates in biological systems. In vitro tests provide essential data to confirm the biological significance of molecular interactions and how compounds may behave in real life. By identifying the most likely interactions among millions of possible interactions, simulations can reduce the cost and time of laboratory studies and help one to understand molecular mechanisms. As a result, these simulations play a vital role in discovering potential drug candidates for treating complex diseases such as AD.

In conclusion, studies conducted in the last five years have evaluated the cholinesterase inhibitory activities of compounds derived from brown algae. These inhibitor activities are generally expressed as IC_50_ values, ranging from 0.0012 to 5 mg mL^−1^. This range varies depending on the algae species, the richness of the extracts in bioactive compounds, and the compound structure and type. AD involves many factors beyond cholinesterase inhibitors. Characteristic features of Alzheimer’s include A*β* plaque formation, tau protein aggregates, oxidative stress, and neuroinflammation. Therefore, evaluating the neuroprotective effects of extracts from brown algae requires a multifaceted approach.

### 4.2. Amyloid-Beta (Aβ) Inhibition

A*β* inhibitors are essential in treating AD because they aim to reduce the production or accumulation of Aβ peptides in the brain. For example, Chagas Monteiro et al. (2023) have stated that inhibiting *β*-secretase (BACE-1) and γ-secretases, which play a role in the degradation of amyloid precursor protein (APP), is effective in preventing the accumulation of A*β* plaques and oligomers [58].

Several metabolites from seaweeds have shown anti-amyloidogenic potentials (Table 2). For example, Kim et al. (2020) have stated that the ethanol extract of *Ishige foliacea* inhibits the BACE1 enzyme dose dependently (effective at the highest concentration (77.8 ± 1.9%, 1000 μg mL^−1^), and the IC_50_ value for BACE1 inhibition was 266.8 ± 34.9 μg mL^−1^ [59].

In one study, 8,8′-bieckol, dieckol, and eckol from *Ecklonia cava* were found to have lower IC_50_ values for BACE1 inhibition than the resveratrol (IC_50_, 14.89 ± 0.54 µM) used as a positive control—1.62 ± 0.14 µM for 8,8′-bieckol, 2.34 ± 0.10 µM for dieckol, and 7.67 ± 0.71 µM for eckol. This study demonstrates the effective inhibition of these compounds on BACE1 [60].

Monoamine oxidases A and B (hMAO-A and hMAO-B) are important isoenzymes that catalyze the oxidative deamination of neurotransmitters and amines and play critical roles in regulating synaptic neurotransmission. However, overexpression of hMAOs can cause oxidation reduction imbalances and mitochondrial damage, leading to neurodegenerative diseases [61]. Kwon et al. (2022) identified two new terpenoid lactones (1 and 2) from *Sargassum macrocarpum* and found that these compounds showed inhibitory activity on hMAO-A. These studies provide significant progress in understanding the potential mechanisms of compounds derived from seaweeds in treating AD [62].

### 4.3. Inhibition of Neurotoxic Effects in Cells

Excessive activation of immune cells such as microglia and astrocytes results in the release of inflammatory mediators such as pro-inflammatory cytokines (e.g., TNF-α, IL-1*β*, and IL-6). These mediators cause increased oxidative stress in brain cells [63]. Oxidative stress, caused by excessive accumulation of oxidizing species, causes imbalances and disruptions in antioxidant defense systems. This process damages the biomolecules of nerve cells, disrupting cell functions and causing cell death and neurotoxicity [64]. Therefore, compounds with antioxidant, anti-inflammatory, anti-amyloidogenic, and anti-aggregation activities are expected to show potential neuroprotective effects. It has been reported that some metabolites isolated from brown seaweeds provide neurological protection against various toxic stimuli thanks to their antioxidant and anti-inflammatory properties.

For instance, The Keap1-Nrf2/HO-1 pathway protects cells against oxidative damage by Keap1 releasing Nrf2 under oxidative stress and Nrf2 traveling to the nucleus and increasing the expression of antioxidant genes. In this sense, Qi et al. (2024) determined that acetoxypachydiol (APHD), a diterpene extracted from brown algae belonging to the *Dictyota* genus, provides activation of the Keap1-Nrf2/HO-1 signaling cascade [65].

The acetone extract of *Padina gymnosperm (ACTPG)* and its active component α-bisabolol showed A*β*-mediated neuroprotective effects. They reduced reactive oxygen and nitrogen species by inhibiting cholinesterase and *β*-secretase enzymes [66].

Compounds from *P. binghamiae* showed antioxidant and anti-apoptotic effects, protecting against glutamate-induced cell damage in HT-22 cells through the Nrf2/HO-1 signaling pathways [43]. In another study, extracts from *Fucus vesiculosus* and *Pelvetia canaliculata* restored cell viability against A*β*25–35-induced toxicity. The *P. canaliculata* extract showed more protective effects over a wider concentration range than the *F. vesiculosus* extract [67].

Olasehinde et al. (2019) examined the neuroprotective effects of seaweeds (*Ecklonia maxima* (KPM), *Gracilaria gracilis* (GCL), *Ulva lactuca* (ULT) and *Gelidium pristoides* (MNP)) against zinc (Zn)-induced neurotoxicity [68]. In the study on HT-22 cells, seaweed extracts pre-applied to cells treated with zinc sulfate positively affected cell viability and inhibited zinc-induced cell death. Additionally, seaweed extracts showed effects on increased apoptosis, oxidative stress markers (malondialdehyde and nitric oxide), and antioxidant enzyme activities (catalase and superoxide dismutase) after Zn treatment.

The neuroprotective effects of six different component fractions of *Ecklonia radiata* were examined in the study [69]. All fractions inhibited A*β*1–42-induced apoptosis and increased neurite outflow, and five fractions reduced A*β*1–42 aggregation and increased the viability of PC-12 cells by reducing hydrogen peroxide (H_2_O_2_)-induced oxidative stress.

The potential role of phlorotannins in AD pathologies has also been highlighted. For example, four phlorotannin compounds (eckol, dieckol, fluorofofuroekol-A (PFFA) and 974-A) obtained from the brown seaweed species *Ecklonia* were examined; PFFA and 974-A protected PC12 cells against H_2_O_2_ and t-BHP-induced oxidative stress and A*β*1–42-induced neurotoxicity. They did not inhibit A*β*1–42 aggregate morphology, indicating that their neuroprotective activity is independent of direct interactions with the A*β*1–42 protein [41,70].

Another research showed eckol, dieckol and phlorofucofuroeckol-A (PFFA) compounds obtained from brown seaweeds to prevent the development of AD pathologies and protect cognitive functions by improving the cytokine-induced intestinal epithelial barrier function and reducing reactive oxygen species [70].

Dieckol (DEK), one of the phlorotannins isolated from *Ecklonia cava*, has been found to strongly reduce intracellular reactive oxygen species (ROS), mitochondrial Ca^2+^ overload, mitochondrial membrane potential (ΔΨm) impairment, and adenosine triphosphate (ATP) depletion through activation of the Nrf2/HO-1 pathway, thus showing a neuroprotective effect against glutamate toxicity [71]. Dibenzodioxin-fucodiphloroethol (DFD) compound isolated from *Ecklonia radiata* prevented the aggregation of A*β*1–42. It also reduced A*β*1–42-induced oxidative stress by suppressing mitochondrial dysfunction and caspase activation and contributing to the down-regulation of pro-inflammatory enzymes through negative regulation of the NF-κB pathway [27].

Barbosa et al. (2020) showed that phlorotannin extracts obtained from *Fucus* species inhibited the activity of monoamine oxidase A and B, tyrosinase, and cholinesterases, enzymes associated with events contributing to the onset and progression of neurodegeneration. Moreover, phlorotannin extracts exhibited multifunctional antioxidant properties that counteract glutamate toxicity in human-derived SH-SY5Y neuron cells [72].

The main phlorotannins from *E. cava*, eckol, dieckol, and 8,8′-bieckol, were found to potently inhibit BACE-1 and AChE enzymes. According to in silico analyses, in the inhibition of BACE-1, eckol established hydrogen bonds with GLY34 and SER36; dieckol with TRP76, THR232, and LYS321; and 8,8′-bieckol with LYS107, GLY230, THR231 and SER325. In the inhibition of AChE, eckol established hydrogen bonds with THR83, TRP86, TYR124, and SER125; dieckol with ASN233, THR238, ARG296, and HIS405; and 8,8′-bieckol with ARG296. Kinetic studies have shown that eckol, dieckol, and 8,8′-bieckol bind to the BACE1 enzyme as a noncompetitive inhibitor and reduce the *Vmax* value without changing the *Km* value between the enzyme and the substrate. These compounds increase the *Km* value without changing the *Vmax* value by binding to the AChE enzyme as a competitive inhibitor [73].

Meshalkina et al. (2023) showed that extracts enriched in intracellular and cell wall-bound phlorotannins isolated from *Fucus vesiculosus* and *Pelvetia canaliculata* had a significant protective effect against A*β*25-35 toxicity. While this effect was effective for *F. vesiculosus* only at high concentrations (5 and 10 μg mL^−1^), *P. canaliculata* extracts were protected at nearly control levels at all tested concentrations [67].

Fucosterol (FST) isolated from *Sargassum horneri* reduced oxidative stress in TNF-α/IFN-γ-stimulated human dermal fibroblasts (HDF) and inflammatory responses in the cells by regulating Nrf2/HO-1 and NF-κB/MAPK signaling pathways [74]. Fucosterol from *Padina australis* showed anti-Alzheimer effects by protecting SH-SY5Y cells against A*β*-induced neurotoxicity, reducing APP mRNA levels and intracellular A*β* levels, and increasing Ngb mRNA levels [75].

Anti-inflammatory compounds have also been investigated for their protective effects on nerve cells. Eleganolone, isolated from *Bifurcaria bifurcata*, reverses 6-OHDA-induced neurotoxicity by approximately 20%. It exhibits neuroprotective effects through mitochondrial protection, oxidative stress reduction, inflammation and apoptosis inhibition, and NF-kB pathway inhibition [76].

Terpenoids are important compounds, naturally obtained from biodiversity-rich sources such as seaweeds. Terpenoids, especially those isolated from seaweeds, exhibit antioxidant, anti-inflammatory, and neuroprotective effects at the cellular level. Qi et al. (2023) isolated five new and 15 known xenican diterpenes from *Dictyota coriacea*. The study determined that all compounds provided significant neuroprotective effects against oxidative stress in PC12 cells and that 18-acetoxy-6,7-epoxy-4-hydroxydictyo-19-al had an antioxidant effect mechanism through the Nrf2/ARE signaling pathway [77].

Diterpene compounds isolated from *Dictyota* sp. showed strong antioxidant effects against H_2_O_2_-induced oxidative damage in neuron-like PC12 cells, and the antioxidant property of dictyol C, one of these compounds, was found to be associated with the activation of the Nrf2/ARE signaling pathway [78].

Kumagai et al. (2024) also isolated seven meroterpenoids from *Dictyopteris polypodioides*. Among these, yahazunol, zonarol, and isozonarol compounds were found to suppress NO production and induce nitric oxide synthase, interleukin-6 and C-C motif chemokine ligand 2 mRNA expression by inhibiting lipopolysaccharide-induced inflammation in RAW264 cells. It has also been revealed that the hydroquinone group in the compounds has a vital role in anti-inflammatory activity [79].

Nine new chroman-type meroterpenoids isolated from *Sargassum siliquastrum*, especially sargasilol A with a shorter carbon chain, showed anti-neuroinflammatory effects by inhibiting LPS-induced NO production. By targeting the IKK/IκB/NF-κB signaling pathway, the compounds blocked the transport of p65 from the cytosol to the nucleus, which reduced the expression of anti-inflammatory cytokines (IL-1*β*, IL-6, TNF-α) [51].

Studies on the effects of fucoidan species on neurological diseases have revealed their protective properties against damage caused by A*β*1–42 and oxidative stress. Xing et al. (2023) found that Type II fucoidan (FvF) obtained from *Fucus vesiculosus* reduced MPP^+^-induced mitochondrial damage and ROS accumulation and prevented neuronal apoptosis [80]. Five different fucoidans with different chemical compositions were obtained from the species *Fucus vesiculosus* (FE, FF, and S) and *Undaria pinnatifida* (UE and UF). Of these, fucoidan S, UE, and UF showed anti-aggregation effects against A*β*1–42, and all fucoidan samples reduced A*β*1–42 and hydrogen peroxide-induced cytotoxicity and inhibited A*β*1–42-induced apoptosis [22].

Subermaniam et al. (2023) have reported that compounds obtained from *Padina australis* (methyl α-eleostearate, ethyl α-eleostearate, niacinamide, stearamide, and linoleic acid) inhibit the excessive production of reactive oxygen species (ROS) and proinflammatory cytokines, suppressing neuroinflammation and microglial activation in central nervous system disorders [81]. Fucoxanthin from *Sargassum oligocystum* Montagne counteracts the neurotoxicity caused by amyloid *β*-protein fragment 25-35 in a C6 neuron cell line by regulating the gene expression of antioxidant enzymes (CAT and GPx) and the PI3K/Akt signaling pathway (GSK-3*β*) and protects against it by increasing the expression of genes involved in autophagy (p62 and ATG5) and acetylcholine biosynthesis (VAChT and ChAT) [20].

In summary, the resulting bioactive compounds can show neuroprotective effects based on their antioxidant, anti-inflammatory, anti-amyloidogenic, and anti-aggregation activities by targeting processes that play an essential role in the pathophysiology of AD. In short, they can reduce oxidative stress and control inflammation by regulating signaling pathways such as NF-κB/MAPK and Nrf2/HO-1. These mechanisms support neuronal health by contributing to the protection of biomolecules in nerve cells. Furthermore, phlorotannin compounds, such as eckol, dieckol, and phlorofucofuroeckol-A, can prevent neurotoxicity and protect cognitive functions in the early stages of AD, primarily by reducing A*β*1–42-induced toxicity and inhibiting apoptosis.

### 4.4. In Vivo Evaluation of Neuroprotective Effects

The neuroprotective properties of some brown algae extracts and compounds obtained from seaweeds are supported by in vivo studies in which various biological activities have been reported (Table 2). For example, Fei et al. (2023) showed that, after oral and intravenous administration of zonarol (20–600 ng mL^−1^) obtained from *Dictyopteris undulata*, this marine hydroquinone is found at a higher level in the brain tissue of mice than in other tissues [82].

Jo et al. (2023) observed that *E. cava* reduced neuroinflammation in mice in the chronic inflammation model caused by lipopolysaccharide (LPS). Mice were pretreated with *E. cava* extract (10–100 mg kg^−1^) orally for 19 days and then exposed to LPS for 1 week. *E. cava* treatment reduced pro-inflammatory cytokine levels in the blood and brains of mice. The study also examined the activation of inflammation-related proteins such as NF-κB and STAT3. *E. cava* treatment was found to regulate brain inflammatory response by reducing NF-κB and STAT3 phosphorylation [83].

Oral administration of ethanol extract of *Ishige foliacea* (250–500 mg kg^−1^) increased the expression of brain-derived neurotrophic factor (BDNF) and tropomyosin receptor kinase B (TrkB)-phosphorylated extracellular signal-regulated kinase protein expression, which is associated with synaptic plasticity in the hippocampus. In vitro tests supported the neuroprotective effects, suggesting that the extract may prevent neurodegenerative disorders [59].

Yende et al. (2021) demonstrated the cognitive-improving effects of the oral administration (200–600 mg kg^−1^) of *Sargassum ilicifolium* (SI) and *Padina tetrastromatica* (PT) extracts containing scopolamine on amnesic mice. The extracts significantly ameliorated scopolamine-induced memory impairments, protecting against antioxidant enzymes and inhibiting the AChE enzyme [84].

In the study of Wang et al. (2022), it was determined that fucoidan (*Fucus vesiculosis*) administered intraperitoneally at 10 mg kg^−1^ showed neuroprotective effects, improving neuroinflammation, promoting neurogenesis, and reducing blood–brain barrier and intestinal barrier permeability in mice with LPS-induced cognitive impairment [85]. A different study evaluated the protective effect of fucoidan isolated from *Sargassum wightii* Greville on streptozotocin-induced cognitive impairments, oxidative stress, and amyloidosis. Streptozotocin-treated animals performed poorly in behavioral assays and exhibited marked cognitive impairments. A significant increase in hippocampal MDA, nitrite, AGE, AChE activity and a decrease in GSH and SOD levels were observed in the disease group. Moreover, behavioral disorders, oxidative stress, hyperphosphorylated tau protein, and amyloidosis symptoms were significantly improved in groups treated with 100 and 200 mg kg^−1^ doses of fucoidan. Histochemical studies further supported the neuroprotective role of fucoidan in the cerebral cortex and hippocampus [86].

Consequently, these bioactive compounds from brown algae are potential candidates for treating neurodegenerative disorders such as AD. Research in this field should continue and be examined in more depth. This study describes the basic steps of in vivo testing and summarizes the potential effects of these compounds based on current research findings.

**Table 2 nutrients-16-04394-t002:** Neuroprotective effects of various brown algae extracts studied in vitro and in vivo.

In vitro					
*Species*	Extraction Approach	Possible Bioactive Compound	Pharmacological Markers/Biological Tests	Neuroprotective Effect	References
*Sargassum Horneri*	Enzyme-assisted extraction	Sulphated fucooligosaccharide	AChE and BChE	AChE (IC_50_: 4.0–14.4 µM) and BChE (IC_50_: 18.5–25.3 µM) inhibition	[47]
*Ecklonia cava*	Ethanol reflux extraction	Polyphenol and fucoidan	Mitochondrial reactive oxygen species (ROS) content amyloid-*β* production; tau hyperphosphorylation-mediated proteins; mitochondrial membrane potential (MMP, ΔΨm); adenosine triphosphate (ATP) content; mitochondria-mediated protein analysis (BAX, cytochrome C)	Reduced AChE activity; reduced mitochondrial ROS; ATP production and MMP restored down-regulating amyloid-*β* production (by JNK/IRS-1/IDE pathway); reduced tau hyperphosphorylation (by PI3K/Akt/GSK-3 pathway)	[8]
*Ecklonia cava*	Ethanol extraction	Phlorotannins	AChE and BChE	AChE (IC_50_: 0.9 ± 0.8 to 66.5 ± 0.4 μM) and BChE (IC_50_: 1.4 ± 3.8 to 25.2 ± 0.1 μM) inhibition	[56]
*Ecklonia cava*	Ethanol extraction	Eckol, dieckol, phlorofucofuroeckol-A (PFFA) and 974-A	H_2_O_2_, t-BHP, A*β*1–42	Effective in ROS scavenging but not in protecting against oxidative stress-evoked neurotoxicity; PFFA and 974-Al; provided broad neuroprotective activity, including protection against oxidative stress and A*β*1–42.	[70]
*Ecklonia cava*	Ethanol extraction	Phlorotanninn and dieckol	AChE and BChE, AAPH and H_2_O_2_-induced oxidative stress in PC-12 and SH-SY5Y cells	AChE Inhibition (95.4%), BChE inhibition (74.7%), reduction in oxidative stress (26.3 to 51.1%)	[41]
*Durvillaea incurvata*	Ultrasound-assisted and conventional extraction	Crude ethanol extract	AChE and BChE	AChE (IC_50_: 48.55 ± 0.021 µg mL^−1^, 51.5% inhibition), BChE (IC_50_: 87.58 ± 0.044 µg mL^−1^, 32.8% inhibition)	[55]
*Petalonia binghamiae*	Sequential maceration	Achanic acid methyl ester, (R)-ricinoleic acid, saikosaponin E, acanthoside D and cansunin D	Glutamate-induced excitotoxicity, ROS scavenging, cell viability	Protection HT-22 cells from glutamate-induced excitotoxicity, increased cell viability and preserved cell morphology, reduced intracellular ROS production and increased HO-1 expression via Nrf2 activation.	[43]
*Sargassum oligocystum Montagne*	Maceration	Fucoxanthin	AChE, cytotoxic effect on C6 cells, neuroprotective effects against H_2_O_2_-induced oxidative stress and A*β*25–35-induced neurotoxicity, gene expression related to antioxidant enzymes (SOD, CAT, GPx), gene expression related to PI3K/Akt signaling (GSK-3*β*), ER stress and apoptosis-related gene expression (CHOP, Bax, caspase-3), gene expression related to ACh biosynthesis (ChAT, VAChT), protein translation (S6K1), autophagy regulation (p62, ATG5)	AChE (IC_50_: 130.12 ± 6.65 μg mL^−1^), protected C6 cells, viability increased to 91.23% at 100 μg mL^−1^ after H_2_O_2_ exposure, Increased cell survival rate significantly from 59.01% to 80.98% at 100 μg mL^−1^ after A*β*25–35 exposure; increased GPx activity by 105.81% at 100 μg mL^−1^ and CAT activity by 31.98% at 50 μg mL^−1^ after H_2_O_2_ exposure; increased mRNA expression of CAT and GPx; reversed the decrease in GSK3*β* induced by A*β*25–35; increased mRNA levels of ChAT and VAChT after A*β*25–35 treatment; inverted effect on S6K1 compared with galantamine; increased ATG5 mRNA levels and reduced p62 mRNA levels	[20]
*Himanthalia elongata (L.)*	Subcritical water extraction	Crude extract	•NO and O_2_^•−^ scavenging activity, AChE and BChE	40% AChE and 40% BChE inhibition, protects against oxidative and nitrosative stresses	[54]
*Eisenia bicyclis (Kjellman)*	Subcritical water extraction	Crude extract	•NO and O_2_^•−^ scavenging activity, AChE and BChE	50% AChE and 40% BChE inhibition, protects against oxidative and nitrosative stresses	[54]
*Sargassum angustifolium*	Extraction with acidic solution	Fucoidan	AChE, cytotoxic effects on NB4 cell line, Alterations in cell proliferation and cell cycle-related gene expression, Bcl-2 gene	AChE (IC_50_:1.20 µg mL^−1^); induction of p53, p21, and pro-apoptotic genes; inhibition of anti-apoptotic Bcl-2 gene	[16]
*Padina tetrastromatica*	Subcritical water hydrolysis	Low molecular weight peptides, flavonoids and phenolic compounds	AChE and α-amylase	AChE inhibition (IC_50_: 17.9 ± 0.1 to 65.9 ± 0.1 mg mL^−1^); α-amylase inhibition (2.4 ± 0.1 to 16.0 ± 0.5%)	[32]
*Ishige foliacea*	Maceration	Crude ethanol extract	AChE, BACE1, ROS scavenging	AChE inhibition (IC_50_: 205.1 μg mL^−1^), BACE1 (IC_50_: 266.8 μg mL^−1),^ reduces H_2_O_2_ and A*β*-induced cell death in SH-SY5Y cells	[59]
*Sargassum macrocarpum*	Maceration	Terpenoid lactones	Human monoamine oxidases A and B	hMAO-A inhibition (42.18 ± 2.68% at 200 μM), any activity against hMAO-B	[62]
*Dictyota coriacea*	Maceration	Acetoxypachydiol	Keap1-Nrf2/HO-1, ROS scavenging	Reduces OGD/R and H_2_O_2_-induced cell death in SH-SY5Y cells, increased the mRNA and protein levels of Nrf2 and HO-1, decreased the protein level of Keap1, promoted the transport of Nrf2 to the cell nucleus	[65]
*Padina gymnospora*	Maceration	Crude extract and α-bisabolol	AChE, BACE1, ROS and RNS, apoptotic gene expression	Inhibition of cholinesterase (ChE) and *β*-secretase (BACE1) activity (specific inhibition values not stated); Reduction of ROS and RNS production; Attenuation of lipid and protein oxidation; Restoration of mitochondrial; Reduced Caspase-3 activation, increased Bcl-2 expression membrane potential	[66]
*Fucus vesiculosus*	Maceration	Intracellular and cell wall-bound phlorotannins	A*β*25–35-induced AD model (SH-SY5Y cells)	Protection at 5 μg mL^−1^ and 10 μg mL^−1^	[67]
*Pelvetia canaliculata*	Maceration	Intracellular and cell wall-bound phlorotannins	A*β*25–35-induced AD model (SH-SY5Y cells)	Protection at all concentrations tested (1–10 µg mL^−1^)	[67]
*Ecklonia maxima*	Maceration	Phloroglucinol, catechin, epicatechin, biochanin A, vulgaxanthin and 7,2,4—trihydoxyisoflavanol	Cell viability; apoptosis (AO/EB staining); SOD, CAT, GSH, MDA, NO, AChE	Increased cell viability in HT-22 cells treated with ZnSO_4_; reduced apoptosis in Zn-treated cells; increased SOD and CAT activities; increased GSH levels; reduced MDA levels; decreased NO levels; reduced AChE activity	[68]
*Gelidium pristoides*	Maceration	Phloroglucinol, catechin, epicatechin, biochanin A, vulgaxanthin and 7,2,4—trihydoxyisoflavanol	Cell viability; apoptosis (AO/EB staining); SOD, CAT, GSH, MDA, NO, AChE	Increased cell viability in HT-22 cells treated with ZnSO_4_; reduced apoptosis in Zn-treated cells; increased SOD and CAT activities; increased GSH levels; reduced MDA levels; decreased NO levels; reduced AChE activity	[68]
*Ecklonia radiata*	Enzyme-assisted extraction	Crude extract (CE), phlorotannin (PT), poly-saccharide (PS), free sugar (FS)	A*β*1–42 aggregation, H_2_O_2_-induced cytotoxicity, cell viability	High neuroprotective activity (viability >92% at 3.125–100 μg mL^−1^); inhibition of A*β*1–42 aggregation; antioxidant activity at 12.5–50 μg mL^−1^; enhanced neurite outgrowth (More than 19%)	[69]
*Ecklonia cava*	Maceration	Dieckol	Cell viability, LDH, morphological assessment, ROS scavenging, mitochondrial function, ATP, mitochondrial membrane potential (ΔΨm), mitochondrial Ca^2+^ and ROS, Nrf2/HO-1	Increased cell viability in primary cortical neurons and HT22 neurons; decreased LDH release indicating reduced cytotoxicity; improved neuronal morphology post-glutamate exposure; decreased intracellular ROS levels in both primary cortical neurons and HT22 cells; protected against glutamate-induced mitochondrial dysfunction; rescued ATP depletion in HT22 neurons; prevented ΔΨm disruption in HT22 neurons; attenuated mitochondrial Ca^2+^ overload in HT22 neurons; reduced mitochondrial ROS levels in HT22 neurons; increased HO-1 expression and Nrf2 nuclear translocation	[71]
*Ecklonia radiata*	Maceration	Dibenzodioxin-fucodiphloroethol	Cell viability, A*β*1–42 toxicity and aggregation, molecular docking, AChE, ROS scavenging	DFD (50 µM) does not induce toxicity in PC-12 cells; Rescued PC-12 cell viability at 1.0 and 1.5 µM A*β*1–42; Reduced A*β*1–42 aggregation; Attenuated ROS levels in PC-12 cells; AChE inhibition (IC_50_ = 41.09 µM); Binded to A*β*1–42 with a docking score of −43.28, forming hydrogen bonds with HIS14 and GLU11; Interacted with AChE with a docking score of −43.48, forming hydrogen bonds and π-π stacking; Binded to A*β*1–42 with a docking score of −43.28, forming hydrogen bonds with HIS14 and GLU11; Binded to A*β*1–42 pentamer with a docking score of −64.01, interacting with multiple residues	[28]
*Sargassum horneri*	Maceration	Fucosterol	Cell viability, ROS scavenging, Nrf2/HO-1, TNF-α/IFN-γ, NF-κB/MAPK	Biocompatible with HDF cells up to 120 μM; decreased in a dose-dependent manner the intracellular ROS production in HDFs; upregulated Nrf2 and HO-1 expression in HDF cells; down-regulated inflammatory mediators in TNF-α/IFN-γ-stimulated HDF cells; reduced phosphorylation of NF-κB and MAPK mediators in a dose-dependent manner; decreased molecules related to connective tissue degradation	[70]
*Bifurcaria bifurcata*	Maceration	Eleganolone	Cell viability, ROS scavenging, mitochondrial function, caspase-3, NF-κB, TNF-α, IL-6, IL-10	Increased cell viability after 6-OHDA treatment; reduced ROS levels and H_2_O_2_ production in SH-SY5Y cells exposed to 6-OHDA; preserved mitochondrial membrane potential (MMP) and ATP levels in SH-SY5Y cells; reduced caspase-3 activity in SH-SY5Y cells exposed to 6-OHDA; inhibited NF-κB p65 translocation in SH-SY5Y cells after 6-OHDA exposure; reduced LPS-induced NO production; decreased TNF-α and IL-6 production in LPS-stimulated RAW 264.7 cells	[76]
*Fucus guiryi*	Maceration	Phlorotannins	AChE, BChE, MAO-A, MAO-B	AChE inhibition (IC_50_ μg DE mL^−1^): 969.51 ± 76.99; BChE inhibition (IC_50_ μg DE mL^−1^): 1065.29 ± 35.35; MAO-A inhibition (IC_50_ μg DE mL^−1^): 168.24 ± 5.40; MAO-B inhibition (IC_50_ μg DE mL^−1^): >500; tyrosinaz inhibition (IC_50_ μg DE mL^−1^): 47.99 ± 0.59	[72]
*Fucus serratus*	Maceration	Phlorotannins	AChE, BChE, MAO-A, MAO-B	AChE inhibition (IC_50_ μg DE mL^−1^): 2709.58 ± 55.25; BChE inhibition (IC_50_ μg DE mL^−1^): 3539.79 ± 109.43; MAO-A inhibition (IC_50_ μg DE mL^−1^): 173.80 ± 25.52; MAO-B inhibition (IC_50_ μg DE mL^−1^): >500; tyrosinaz Inhibition (IC_50_ μg DE mL^−1^): 47.66 ± 2.84	[72]
*Fucus spiralis* L.	Maceration	Phlorotannins	AChE, BChE, MAO-A, MAO-B	AChE inhibition (IC_50_ μg DE mL^−1^): >5000; BChE inhibition (IC_50_ μg DE mL^−1^): >5000; MAO-A inhibition (IC_50_ μg DE mL^−1^): 1929.65 ± 100.44; MAO-B inhibition (IC_50_ μg DE mL^−1^): >500; tyrosinaz inhibition (IC_50_ μg DE mL^−1^): 861.73 ± 7.37	[72]
*Fucus vesiculosus* L.	Maceration	Phlorotannins	AChE, BChE, MAO-A, MAO-B	AChE inhibition (IC_50_ μg DE mL^−1^): >5000; BChE inhibition (IC_50_ μg DE mL^−1^): >5000; MAO-A inhibition (IC_50_ μg DE mL^−1^): >3000; MAO-B inhibition (IC_50_ μg DE mL^−1^): >500; tyrosinaz inhibition (IC_50_ μg DE mL^−1^): 2546.82 ± 98.00	[72]
*Padina australis*	Maceration	Fucosterol	Cell viability, apoptosis, APP mRNA, intracellular A*β*, BACE1, ROS scavenging	Increased cell viability in SH-SY5Y cells treated with 2 μM A*β*; reduced apoptosis (30.93%); reduced APP mRNA levels; reduced intracellular A*β* levels; noncompetitive inhibitor on *β*-secretase (BACE1); reduced oxidative stress by increasing anti-oxidant enzyme activities	[75]
*Dictyota coriacea*	Maceration	Xenicane diterpenes	Cell viability, LDH, Nrf2/ARE	Increased cell viability in PC12 cells damaged by H_2_O_2_; reduced LDH levels released from PC12 cells; increased Nrf2 and HO-1 expression in PC12 cells; promoted nuclear translocation of Nrf2	[77]
*Dictyota* spp.	Maceration	Hydroazulene diterpenes	Cell viability, Nrf2/ARE	Increased cell viability in H_2_O_2_-damaged PC12 cells at a concentration of 2 μM; antioxidant effect by activating the Nrf2/ARE signaling pathway	[78]
*Dictyopteris polypodioides*	Maceration	Meroterpenoids: yahazunol (1), zonarol (2), isozonarol (3), and four other meroterpenoids	NO production, iNOS, IL-6, and CCL2 mRNA expression, structure activity of compounds	Inhibited NO production in lipopolysaccharide (LPS)-stimulated RAW264 cells; inhibited iNOS, IL-6, and CCL2 mRNA expression. The hydroquinone unit is important in the anti-inflammatory activity of these sesquiterpenoids	[79]
*Sargassum siliquastrum*	Maceration	Meroterpenoids	BV-2 microglial cells, anti-inflammatory cytokines	Reduced the expression of anti-inflammatory cytokines (IL-1*β*, IL-6, TNF-α); inhibited LPS-induced NO production in BV-2 microglial cells by targeting IKK/IκB/NF-κB pathways	[51]
*Fucus vesiculosus*	Maceration	Fucoidan	CCK-8 (cell viability), LDH, Hoechst staining, MAP2 immunostaining, MitoSOX staining (for ROS), protein target identification (ATP5F1a)	Enhanced cell viability at concentrations of 5, 10, and 25 μM in MPP^+^-induced SH-SY5Y cells; reduced LDH release in MPP^+^-induced SH-SY5Y cells and primary neurons, indicating reduced cell damage. Decreased the proportion of apoptotic cells in MPP^+^-treated primary neurons; protected neurons from MPP^+^-induced axon loss and damage. Reduced mitochondrial ROS production in MPP^+^-treated SH-SY5Y cells. ATP5F1a knockdown reversed the neuroprotective effects of FvF, confirming its role in mitigating mitochondrial dysfunction and apoptosis	[80]
*Fucus vesiculosus*	Maceration	Fucoidan	A*β*1–42 aggregation and cytotoxicity, ROS Scavenging, neurite outgrowth	Protected against A*β*1–42-induced cytotoxicity; inhibited A*β*1–42 aggregation; slight protection against hydrogen peroxide-induced cytotoxicity; inhibited apoptosis induced by A*β*1–42	[22]
*Padina australis*	Maceration	Crude ethanol extract	Cell viability, NO, prostaglandin E2, ROS scavenging, iNOS and COX-2 expression, TNF-α and IL-6 secretion	No cytotoxicity at concentrations of 0.25–2.0 mg mL^−1^, reduced at higher concentrations (4.0–8.0 mg mL^−1^); reduced NO production compared with LPS-stimulated levels dose-dependently (0.5–2.0 mg mL^−1^); suppressed PGE2 production at concentrations of 0.5–1.0 mg mL^−1^, increased PGE2 levels at higher concentration (2.0 mg mL^−1^); reduced intracellular ROS generation compared with LPS-stimulated levels dose-dependently; down-regulated iNOS and COX-2 expression induced by LPS; inhibited TNF-α and IL-6 secretion compared with LPS-stimulated levels	[81]
**In vivo**					
** *Species* **	**Possible Bioactive Compound**		**Pharmacological Markers/Biological Tests**	**Neuroprotective Effect**	**References**
*Dictyopteris undulata*	Zonarol		Brain tissue distribution after administration	Higher level in brain tissue than in other tissues	[82]
*Ecklonia cava*	Crude water extracts		Reduction of pro-inflammatory cytokines, NF-κB, STAT3 phosphorylation	Reduced neuroinflammation in LPS-induced model	[83]
*Ishige foliacea*	Crude ethanol extract		BDNF and TrkB expression, synaptic plasticity	Increased BDNF, TrkB-phosphorylated ERK expression	[59]
*Sargassum fusiformis*	Fucoxanthin (Fx)		Reduction of NO, ROS, cell death	Reduced inflammatory responses	[9]
*Sargassum fusiformis*	Phlorotannin and fucoidan mixture		Protection against A*β*-induced learning and memory impairments	Anti-amnesiac effect	[9]
*Sargassum ilicifolium* and *Padina tetrastromatica*	Crude chloroform and ethanol extracts		Improvement of memory impairments, oxidative enzyme protection, AChE inhibition	Cognitive improvement	[84]
*Fucus vesiculosus*	Fucoidan		Improvement of neuroinflammation, promotion of neurogenesis, reduced blood–brain and intestinal barrier permeability	Neuroprotective effects	[85]
*Sargassum wightii Greville*	Fucoidan		Behavioral assays, oxidative stress markers, hyperphosphorylated tau protein, amyloidosis symptoms	Improvement of cognitive impairments and oxidative stress	[86]
*Padina gymnospora*	Crude extract and α-bisabolol		Lifespan in CL2006 and CL4176 mutants	Lowering the expression of AD-related genes (ace-1, hsp-4) in *C. Elegans*	[66]
*Dictyota coriacea*	Xenicane diterpenes		CIRI Model	Reduced brain infarct size and neurological deficit score in the transient middle cerebral artery occlusion (MCAO) rat model	[83]
*Dictyota* spp.	Hydroazulene diterpenes		CIRI Model	Reduction in both brain infarct size and neurological deficit score in transient middle cerebral artery occlusion (MCAO) rat model	[78]
*Fucus vesiculosus*	Fucoidan (FvF)		MPTP mouse model	Improved motor coordination and balance; slowed loss of TH markers; mitigated dopamine neuron loss in the substantia nigra pars compacta (SNpc).	[80]
*Laminaria japonica*	Fucoidan		Adenine-induced CKD mice	Reduced signs associated with cognitive behavior; improvement in genes related to Alzheimer’s disease and memory; regulated of inflammatory response via microglia/macrophage polarization; ameliorated oxidative stress via Nrf2-HO-1 signaling pathway; improved cognitive impairments (new object recognition, object location, passive avoidance tests)	[87]
*Fucus vesiculosus*	Fucoidan		Mongolian gerbils (*Meriones unguiculatus*)	Reduced tGCI-induced hyperactivity; reduced the loss of NeuN^+^ pyramidal cells in the CA1 region after tGCI; reduced the number of F-J B^+^ cells in the CA1 region; reduced the activation of GFAP^+^ astroglia and the production of reactive oxygen species; reduced the activation of Iba-1^+^ microglia and the production of reactive oxygen species; reduced HNE immunoreactivity and lipid peroxidation in CA1 pyramidal cells; reduced superoxide levels in CA1 pyramidal cells; increased SOD1 and SOD2 expression in CA1 pyramidal cells	[88]
*Laminaria digitata*	Laminarin		Mongolian gerbils	Decreased superoxide anions; decreased IL-1*β* and TNF-α; increased SOD expression; increased IL-4 and IL-13 expression	[89]
*Petalonia binghamiae*	Water extract		MCAO/R mice model	Decreased neuronal death in ischemic lesion; attenuated sensorimotor deficits; reduced ROS levels; decreased apoptosis in ischemic area; enhanced Nrf2 nuclear translocation; increased HO-1 expression	[43]
*Sargassum boveanum*	Crude ethanolic extract		Sprague–Dawley rats	Increased antioxidant enzyme activities (SOD, GPx) and gene expressions (SOD, GPx, Nrf2, HO-1); decreased expression of antioxidant genes (SOD, GPx, Nrf2, HO-1); increased expression of inflammatory markers (TNF-α, NF-κB)	[90]
*Sargassum angustifolium*	Hydroalcoholic, methanolic, and hexane extract		Reversal of memory impairment, passive avoidance test, Morris water maze test	Improvement of cognitive functions	[91]

## 5. Conclusions

This narrative review systematically examined the breadth of the existing literature on bioactive compounds from brown macroalgae and identified knowledge gaps in this area. Furthermore, the potential of brown macroalgae for bioactive compounds with neuroprotective properties was evaluated. In conclusion, compounds obtained from brown algae, especially polysaccharides, phenolic compounds, and omega-3 fatty acids, have antioxidant, anti-inflammatory, and neuroprotective effects. Advanced extraction methods, especially supercritical fluid extraction (SFE) and ultrasound-assisted extraction (UAE) offer significant advantages in increasing the isolation efficiency of these valuable compounds. Furthermore, the wide range of neuroprotective compounds requires the development of advanced analytical methods due to the structural diversity and complexity of the extracts. In this context, HPLC is the most frequently used technique. Nevertheless, HPLC and GC should be used in tandem with off-target (Q-TOF) and on-target (triple quadrupole) or NMR to obtain an in-depth chemical characterization of these natural compounds. This will allow a comprehensive evaluation of the efficiency and the biological activities of the obtained bioactive compounds. To fully evaluate the true potential of the brown algae extracts, in vitro studies are needed to understand the mechanisms, and in vivo experiments are required to better realize their predictable effects on humans. In this way, it will be possible to understand the mechanisms of action of existing compounds on neuroprotection, particularly against AD, before their potential use as functional foods and dietary supplements. However, more research is needed to explore brown algae’s potential in nutrition and health applications.

## Figures and Tables

**Figure 1 nutrients-16-04394-f001:**
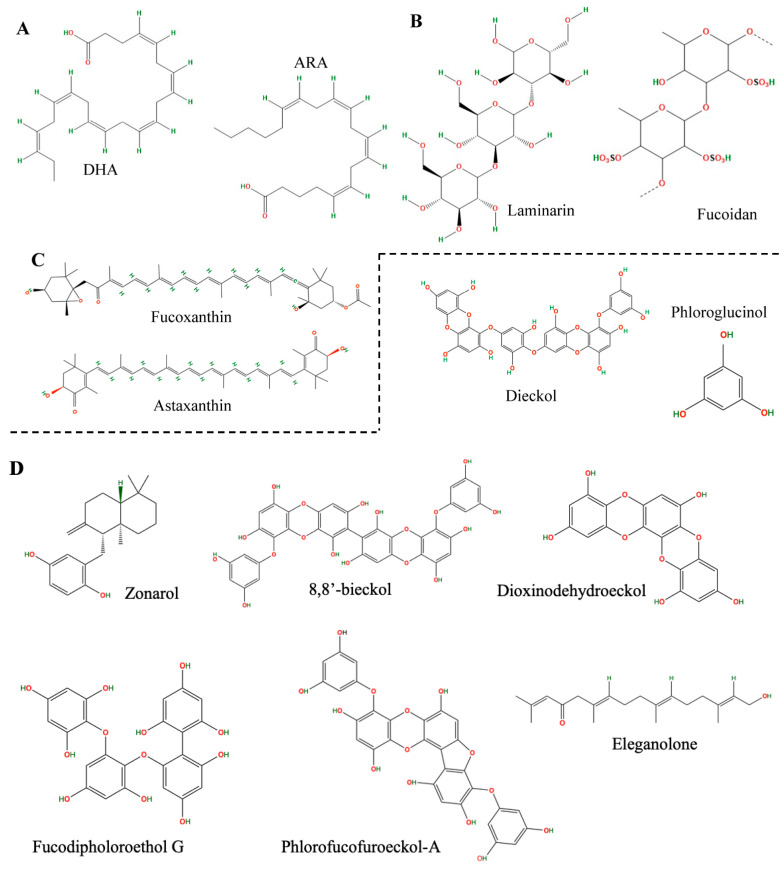
Chemical structure of some neuroprotective compounds from brown macroalgae. (**A**) Omega-3 fatty acids: Docosahexaenoic acid (DHA) and arachidonic acid (ARA). (**B**) Polysaccharides: Laminarin and fucoidan. (**C**) Carotenoids: Fucoxanthin and astaxanthin. (**D**) Polyphenols and their derivatives: Dieckol, phloroglucinol, zonarol, dioxynodehydroeckol, eleganolone, fluorofucofuroeccol-A, fucodifluoroetol G and 8,8′-bieckol.

**Figure 2 nutrients-16-04394-f002:**
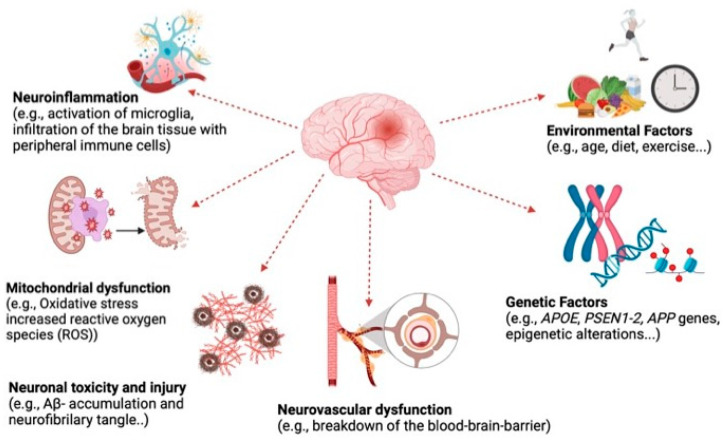
Fundamental mechanisms that cause neurodegeneration (created by BioRender.com).

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
