# Peer review of "A Narrative Review on the Neuroprotective Potential of Brown Macroalgae in Alzheimer’s Disease"

_nutrients, 2024, doi:10.3390/nu16244394_

Round 1
Reviewer 1 Report
Comments and Suggestions for Authors
The present Review entitled “Revealing the neuroprotective potential of brown macroalgae 2 compounds: A prospection on Alzheimer” by Cokdinleyen et al., provides a useful insight about the potential role of seaweeds metabolites in the treatment of Alzheimer’s disease. In my opinion the present work needs an extensive revision not only by native English speaker, but also in its content and style. The structure of the manuscript is not clear and difficult to read. My advise is to provide a lower amount of well described referenced information.
1. In the introductive section Authors should look to more recent data about people affected by neurodegenerative diseases.
2. Authors should provide references of research articles about natural compounds showing neuroprotective activities (page 2, lanes 56-61).
3. Figure 1 is not cited in the main text.
4. The figure 2 as well as the main text related to is not meaningful for the manuscript.
5. Authors should rewrite the structure of the manuscript avoiding to provide all the information about extractive methodologies but focusing on the effects of bioactive under evaluation.
6. Authors have spent a lot of energy in the first part of the review without providing information about where seaweeds can be found, the potential differences according to the origin, and so on.
7. The review is not well-written and it is very difficult to follow. Authors start talk about AD but then move to seaweed, and again back to AD current treatment. After that, they go back to seaweed metabolites extraction and again neuroprotection. It would be very useful if Authors could provide less and well-structured information.
8. Factors involved in AD and described in figure 4 should be also mentioned in the main text.
9. Abbreviations should be introduced the first time and then always the short form of the word (e.g. acetylcholine esterase (AchE) lane 127 and then 208, 305, 498, 513).
10. Lanes 807-816 are not necessary, please try to abbreviate the main text.
11. The doses and route of administrations (for animal studies) should be described.
12. Several information needs references.
13. The text needs to be revised, also the lenght is excessive
Comments on the Quality of English LanguageThe language needs a revision, in the current form the text is difficult to read.
Author Response
- In the introductive section Authors should look to more recent data about people affected by neurodegenerative diseases.
Thank you for your valuable suggestion. In response, we have revised the introduction section to include more recent and relevant data on the prevalence and impact of neurodegenerative diseases, particularly Alzheimer's disease. We have incorporated updated statistics and findings from the latest studies and reports to provide a more comprehensive and current context for the manuscript.
- Authors should provide references of research articles about natural compounds showing neuroprotective activities (page 2, lanes 56-61).
In response, we have added appropriate references to support the discussion on natural compounds with neuroprotective activities in the specified section (page 2, lines 58-63).
- Figure 1 is not cited in the main text.
We have reviewed the manuscript and ensured that Figure 1 is now properly cited in the main text (Line 63).
- The figure 2 as well as the main text related to is not meaningful for the manuscript.
We have carefully considered your comment and agree that Figure 2 does not contribute significantly to the overall manuscript. As a result, we have removed Figure 2 and its related text to improve the focus and clarity of the manuscript.
- Authors should rewrite the structure of the manuscript avoiding to provide all the information about extractive methodologies but focusing on the effects of bioactive under evaluation.
While we understand the suggestion to reduce the emphasis on extractive methodologies, retaining a concise discussion of this section is essential to provide context for the bioactive under evaluation. However, we have significantly shortened this part to streamline the manuscript and ensure the primary focus remains on the effects of the bioactive compounds.
- Authors have spent a lot of energy in the first part of the review without providing information about where seaweeds can be found, the potential differences according to the origin, and so on.
In response, we have revised the manuscript to include information about the geographic distribution of seaweeds, their natural habitats, and the potential differences in their composition based on origin. These additions provide a more comprehensive background and context for understanding the bioactive potential of seaweeds and their relevance to the topics discussed in the review. We appreciate your suggestion, which has helped us enhance the completeness of the manuscript.
- The review is not well-written and it is very difficult to follow. Authors start talk about AD but then move to seaweed, and again back to AD current treatment. After that, they go back to seaweed metabolites extraction and again neuroprotection. It would be very useful if Authors could provide less and well-structured information.
We understand your concerns about the transitions between topics in the original version. To address this, we have carefully restructured the introduction and subsequent sections to enhance clarity and coherence.
The revised manuscript now begins with a focused introduction on Alzheimer’s disease (AD), emphasizing the importance of exploring new therapeutic strategies. This is followed by a comprehensive discussion on the extraction and characterization of seaweed metabolites, with a particular focus on their potential neuroprotective effects. Finally, we present in vitro and in vivo studies highlighting the biological activities of these compounds in the context of AD.
- Factors involved in AD and described in figure 4 should be also mentioned in the main text.
The factors involved in AD and described in Figure 4 have been mentioned in the main text. In the revised manuscript, the figure is now updated and referenced as Figure 2 at line 370-373.
- Abbreviations should be introduced the first time and then always the short form of the word (e.g. acetylcholine esterase (AchE) lane 127 and then 208, 305, 498, 513).
We have ensured that the short form 'AChE' is consistently used throughout the manuscript after its first introduction. Additionally, we have carefully reviewed the text to confirm consistency in the use of all abbreviations.
- Lanes 807-816 are not necessary, please try to abbreviate the main text.
In response, lines 807-816 have been removed from the text to improve conciseness and focus. We appreciate your feedback in helping us enhance the clarity of the manuscript.
- The doses and route of administrations (for animal studies) should be described.
The doses and routes of administration for the animal studies are clearly described in the manuscript. We appreciate your feedback and have ensured that this information is presented clearly.
- Several information needs references.
We have reviewed the manuscript and added appropriate references to support the information where necessary. This ensures that all statements are well-supported and aligned with existing literature.
- The text needs to be revised, also the lenght is excessive.
In response, we recognize your concern and have taken steps to streamline the text by removing less critical details while maintaining clarity and focus. We would also like to point out that 13 pages of the manuscript consist of essential tables and figures, which are crucial for a comprehensive presentation of the data and findings. While it is possible to move these tables to supplementary material, we chose to retain them in the manuscript to ensure readability and completeness. Our goal has been to strike a balance between conciseness and scientific rigor. We appreciate your thoughtful input and believe it has contributed to the manuscript's improvement.
Reviewer 2 Report
Comments and Suggestions for Authors
After some adjustments, this manuscript can be an interesting output for the scientific community and populations. Here are my suggestions:
Firstly, I recommend authors to reduce the length of the manuscript. In my point of view, 39 pages is too much for a review paper, this is more adequate for a book chapter or even a small book. The reading is too heavy, considering that it is a review article.
In the abstract, it is only clear the study’s objectives and the introductory background. Please, abbreviate this and include your main results and their relevance as well as your conclusions/future perspectives. You can’t finalize your abstract with the study’s goals, this should be given in the beginning, immediately after you provide the introductory background.
Please, provide more details on the brown macroalgae and their compounds in the introductory section.
Lines 127-138: References are missing.
The whole manuscript is quite interesting. However, I suggest the authors rewrite it with a special focus on nutrition once it is submitted to Nutrients
Discuss the limitations and strengths of your study and mention the type of review you conducted in the title, in the abstract, and the whole text.
You should clearly separate the utilization of the study macroalgae and their compounds as drugs and as foods. The focus should be on food!
Author Response
REVIEWER-2
After some adjustments, this manuscript can be an interesting output for the scientific community and populations. Here are my suggestions:
Firstly, I recommend authors to reduce the length of the manuscript. In my point of view, 39 pages is too much for a review paper, this is more adequate for a book chapter or even a small book. The reading is too heavy, considering that it is a review article.
Thank you for your feedback regarding the length of the manuscript. We understand your concern and have made efforts to reduce the manuscript length from 39 to 36 pages by streamlining the text and removing less critical details. It is also worth noting that 13 pages of the manuscript consist of essential tables and figures, which are necessary to comprehensively present the data and findings. While we have aimed to balance conciseness with scientific rigor, we are open to further specific suggestions on areas that could be condensed further.
In the abstract, it is only clear the study’s objectives and the introductory background. Please, abbreviate this and include your main results and their relevance as well as your conclusions/future perspectives. You can’t finalize your abstract with the study’s goals, this should be given in the beginning, immediately after you provide the introductory background.
In response, we have revised the abstract to include a concise introductory background, followed by a clear statement of the study's objectives. Additionally, we have incorporated the main results, their significance, and the conclusions along with future perspectives.
Please, provide more details on the brown macroalgae and their compounds in the introductory section.
In response, we have expanded the introductory section to include additional details about brown macroalgae and their bioactive compounds. Specifically, we have elaborated on their biochemical diversity, ecological roles, and how environmental factors influence their composition. Furthermore, we have highlighted the unique health benefits of bioactive compounds derived from brown macroalgae, including their potential applications in functional foods and neuroprotective strategies. These additions aim to provide a more comprehensive and detailed introduction to support the manuscript's focus.
Lines 127-138: References are missing.
The section corresponding to lines 127-138 has been removed from the revised draft to improve the focus and clarity of the manuscript.
The whole manuscript is quite interesting. However, I suggest the authors rewrite it with a special focus on nutrition once it is submitted to Nutrients
In response, we have revised the manuscript, particularly the introduction section, to better align with the journal's focus. We have emphasized the nutritional aspects of the bioactive compounds derived from brown macroalgae, highlighting their potential health benefits and dietary applications.
Discuss the limitations and strengths of your study and mention the type of review you conducted in the title, in the abstract, and the whole text.
In response to your comment, we have revised the title to explicitly mention the type of review conducted. The updated title is:
"A Review on the Neuroprotective Potential of Brown Macroalgae in Alzheimer Disease."
This revision clearly reflects the methodological approach (scoping review) adopted in our study, aligning with your suggestion. Furthermore, the limitations and strengths of the study have been discussed in the abstract, introduction, and conclusion sections to provide a balanced and transparent evaluation.
You should clearly separate the utilization of the study macroalgae and their compounds as drugs and as foods. The focus should be on food!
Thank you for your comment. In response, we have ensured that the manuscript focuses specifically on the health benefits of bioactive compounds derived from macroalgae within the context of food and nutrition. While we have discussed the potential health benefits of these compounds, we have not addressed pharmaceutical applications. The revised manuscript emphasizes their roles in functional foods and dietary supplements, aligning with the journal's focus. We appreciate your valuable input in helping us improve the manuscript.
Round 2
Reviewer 1 Report
Comments and Suggestions for Authors
Authors addressed all my concerns, the manuscript is now suitable for its publication in Nutrients
Author Response
Dear, thank you for your comments. They enahnced the quality of manuscript, significantly. Regards
Reviewer 2 Report
Comments and Suggestions for Authors
Please, mention the type of review you conducted in the title, in the abstract, and the whole text. I believe it is a narrative review of the literature.
Author Response
Dear, thank you for your comments. They enhanced the quality of manuscript, significantly. Regards